# Treatment of Periodontal Inflammation in Diabetic Rats with IL-1ra Thermosensitive Hydrogel

**DOI:** 10.3390/ijms232213939

**Published:** 2022-11-11

**Authors:** Yue Liu, Chang Liu, Chang Wang, Qian Zhang, Xingyuan Qu, Chen Liang, Chao Si, Lei Wang

**Affiliations:** 1Department of Periodontology, Hospital of Stomatology, Jilin University, 1500 Tsinghua Road, Chaoyang District, Changchun 130021, China; 2Department of Prosthodontics, Hospital of Stomatology, Jilin University, Changchun 130021, China; 3Department of Orthodontics, Hospital of Stomatology, Jilin University, Changchun 130021, China

**Keywords:** periodontitis, diabetes, injectable and thermosensitive hydrogel, interleukin 1 receptor antagonist protein

## Abstract

Periodontitis is a chronic inflammatory disease that is considered to be the main cause of adult tooth loss. Diabetes mellitus (DM) has a bidirectional relationship with periodontitis. Interleukin-1β (IL-1β) is an important pre-inflammatory factor, which participates in the pathophysiological process of periodontitis and diabetes. The interleukin-1 receptor antagonist (IL-1ra) is a natural inhibitor of IL-1, and the balance between IL-1ra and IL-1β is one of the main factors affecting chronic periodontitis (CP) and diabetes. The purpose of this study is to develop a drug carrier that is safe and nontoxic and can effectively release IL-1ra, which can effectively slow down the inflammation of periodontal tissues with diabetes, and explore the possibility of lowering the blood sugar of this drug carrier. Therefore, in this experiment, a temperature-sensitive hydrogel loaded with IL-1ra was prepared and characterized, and its anti-inflammatory effect in high-sugar environments in vivo and in vitro was evaluated. The results showed that the hydrogel could gel after 5 min at 37 °C, the pore size was 5–70 μm, and the cumulative release of IL-1ra reached 83.23% on the 21st day. Real-time polymerase chain reaction (qRT-PCR) showed that the expression of IL-1β, Interleukin 6 (IL-6), and tumor necrosis factor α (TNF-α) inflammatory factors decreased after the treatment with IL-1ra-loaded thermosensitive hydrogel. Histological evaluation and micro-computed tomography (Micro-CT) showed that IL-1ra-loaded thermosensitive hydrogel could effectively inhibit periodontal inflammation and reduce alveolar bone absorption in rats with diabetic periodontitis. It is worth mentioning that this hydrogel also plays a role in relieving hyperglycemia. Therefore, the temperature-sensitive hydrogel loaded with IL-1ra may be an effective method to treat periodontitis with diabetes.

## 1. Introduction

Periodontitis is a chronic inflammatory and destructive disease of periodontal tissue. With the occurrence and development of the disease, gingival bleeding, periodontal pocket formation, and periodontal purulent loose teeth may occur; thus, it is considered the main reason for adult tooth loss [1]. However, the adverse effects of periodontitis are not limited to periodontal supporting tissues. Persistent periodontitis can affect the overall health of the individual and increase the risk of systemic chronic diseases such as diabetes. Indeed, some scholars have found that the incidence of periodontitis in diabetic patients is 2–3 times higher than that in healthy patients [2,3]. Poor blood glucose control increases the severity of gingivitis and periodontitis, and, as such, periodontitis is considered the sixth complication of diabetes [4]. There is a bidirectional relationship between the two chronic diseases, and both have the ability to induce inflammatory responses. Diabetes can promote apoptosis of periodontal membrane fibroblasts and hinder the repair of periodontal tissue. The inflammatory factors produced by periodontitis can cause circulatory inflammation and affect the blood glucose control and the occurrence and development of complications of diabetes [5]. As one of the systemic risk factors of periodontitis, diabetes not only increases the risk and severity of periodontitis but also affects its treatment and prognosis.

IL-1β is an important multifunctional cytokine, which is closely related to the cause and process of periodontitis and diabetes. In the process of periodontitis, IL-1β promotes fibroblasts to secrete collagenase, interstitial lytic enzyme, and gelatin-degrading enzyme, which leads to matrix degradation, loss of connective tissue, and destruction of periodontal tissue. IL-1β can also stimulate osteoblasts to secrete plasmin and prostaglandin E2 and promote the differentiation of periodontal ligament mesenchymal stem cells into osteoclasts, which leads to increased inflammation and bone absorption [6]. IL-1ra can block the biological activity of IL-1β by binding to the IL-1 receptor, which reduces the expression of inflammatory factors, such as IL-6 and TNF-α, and inhibits the formation of osteoclasts [7]. During the occurrence and development of diabetes, elevated levels of glucose and free fatty acids (FFAs) stimulate islet β cells to produce IL-1β [8]. IL-1β may accelerate its own production by activating Interleukin 1 receptor 1 (IL-1RI), which induces the production of IL-1-dependent cytokines and chemokines [8,9]. Studies have shown that under the stimulation of cytokines such as IL-1β, TNF-α, and interferon γ (IFN-γ), inducible nitric oxide synthase (iNOS), which is seldom expressed in the physiological state, can be produced in large quantities and catalyze the synthesis of excessive nitric oxide (NO). High concentrations of NO can be transformed into free radicals with strong cytotoxicity, causing serious damage to β cells and inhibiting insulin secretion [9]. The mechanisms underlying the protective effect of IL-1ra on pancreatic islets involve its ability to block the stimulation and production of IL-1β and IL-1-dependent cytokines and chemokines induced by glucose, reduce the activity of iNOS in pancreatic islets, prevent excessive production of NO, reduce the cytokine-induced damage to pancreatic islets, and improve the survival and function of pancreatic islets [9]. However, IL-1ra is a water-soluble protein drug, which has low bioavailability and a short half-life, requiring large doses and multiple injections, all of which will reduce its therapeutic effect [10]. Therefore, how to reduce the frequency of administration and improve the efficacy has become a major problem.

The controlled release of IL-1ra is ideal to improve the active duration of the drug and maintain the drug concentration in the local periodontal pocket within the effective concentration range. Temperature-sensitive hydrogel can undergo solution-gel phase transition with a change in temperature, and, as a result, has attracted much attention as an injectable biomaterial with excellent properties. Hydrogel is an insoluble polymer with a three-dimensional network structure, which swells in water and stores high volumes of water. Given the volume of water in the hydrogel stereoscopic network structure, the molecular chain can be extended at will, and it has good fluid properties while maintaining its shape. Moreover, as the biochemical properties of hydrogel are similar to those of human tissue, it has good biocompatibility and is widely used in the field of biomedicine [11,12,13]. The purpose of this study is to develop a new injectable thermo-sensitive hydrogel loaded with IL-1ra for the treatment of periodontal inflammation with diabetes, and to verify the anti-inflammatory properties of the hydrogel through cell experiments and animal experiments, and further explore its effect on blood sugar level. The above contents are marked in red in the revised manuscript.

In our research group, chitosan (CS), β-glycerophosphate (β-GP), and gelatin (gel) were mixed at room temperature, and the hydrogel could be changed from the solution to the gel state in approximately 5 min at 37 °C; this duration could be adjusted by changing the ratio of each component to meet the needs of different treatment sites. Moreover, implantation of temperature-sensitive hydrogel into the body for local treatment does not require surgery. The hydrogel should be injected targeting the lesion site through a syringe, thus minimizing the surgical trauma and improving patient comfort. Chitosan-β-glycerophosphate-gelatin temperature-sensitive hydrogel systems can realize drug-sustained release, thus effectively solving the low bioavailability and short half-life of IL-1ra. In previous experiments, we have discussed the effect of IL-1ra on rats with simple periodontitis, so this experiment will not verify it.

## 2. Results

### 2.1. Synthesis and Characterization of Hydrogels

At room temperature, the IL-1ra CS/β-GP/Gel mixed solution was transparent and clear and changed from a solution state to a uniform gel state after 5 min at 37 °C (Figure 1a). The SEM results showed that the hydrogel was loose and porous, with a pore size of 5–70 μm (Figure 1b).

The CS/β-GP/Gel hydrogel and the external spectra of each group are shown in Figure 1c. The CS at 3113.4 cm^−1^ is the stretching vibration absorption band of -OH and -NH2. The intermolecular and intramolecular hydrogen bonding of CS makes the band wider. The stretching vibration absorption peak of aliphatic C-H is 2952.1 cm^−1^, and the characteristic absorption peak of the chitosan amino group is 1574.3 cm^−1^. The absorption peaks at 1398.4 cm^−1^ and 1339.7 cm^−1^ are the deformation absorption peaks of =CH_2_. The C-H bending vibration absorption peaks were found at 1275.2 cm^−1^ and 1204.9 cm^−1^. The absorption peak at 1131.6 cm^−1^ is the C-O absorption peak on the six-membered ring, while the absorption peak at 1043.6 cm^−1^ is the variable-angle vibration absorption peak of the alcohol hydroxyl group. The analysis of the infrared spectrum of β-GP demonstrated that the absorption peak of the asymmetric stretching vibration of the phosphate radical is at 1002.0 cm^−1^, the symmetrical stretching vibration peak of the phosphate radical is at 921.3 cm^−1^, and the in-plane bending vibration of phosphate radical is at 500–700 cm^−1^. In the infrared spectrum of gelatin, 3025.4 cm^−1^ is the frequency double absorption of the amide II band, 2914.0 cm^−1^ is C-H stretching vibration peak, 1586.0 cm^−1^ is the C-N stretching or N-H bending, and 1483.4 cm^−1^ is caused by -CH_2_- or CH3- bending vibration. The symmetric vibration of the carboxyl group is 1421.8 cm^−1^, the oscillating vibration of CH_3_- is 1357.3 cm^−1^, the C-N stretching or N-H bending is 1292.8 cm^−1^, and the C-O stretching peak is 1172.6 cm^−1^.

Figure 1d shows the X-ray diffraction patterns of chitosan, gelatin, and the CS/β-GP/Gel hydrogel. The results show that chitosan has two characteristic peaks at 2θ = 12.1° and 23.5°, while gelatin has two obvious diffraction peaks at 2θ = 20.9°. After chitosan, gelatin, and β-glycerophosphate were mixed to make the gel, the diffraction peaks of chitosan at 23.5° and gelatin at 20.9° were obviously weakened, and many new diffraction peaks were formed.

Figure 1e shows that the degradation rate of IL-1ra CS/β-GP/Gel hydrogel is fast in the first week, and gradually slows down in the next few weeks, and the degradation rate reaches 75.20% in the eighth week. The drug release curve (Figure 1f) showed that there was a sudden release of IL-1ra in the first three days, and the cumulative drug release percentage reached 51.77% on the third day. The release rate of hydrogel slowed down on the fifth to tenth days, and reached 71.47% on the tenth day. After ten days, the hydrogel released gradually and the drug release amount was less, and on the 21st day the drug release amount was 83.23%.

### 2.2. IL-1ra CS/β-GP/Gel Hydrogels Have Good Biocompatibility

Previous experimental results show that 10 μg/mL IL-1ra can inhibit the release of inflammatory factors in a simulated in vitro inflammatory environment. In this experiment, the dose of IL-1ra is much higher than the required dose. The results of the cell experiment showed that the cell survival rate of RAW264.7 cells cultured with high sugar extract loaded with IL-Ra for 24 and 48 h was not significantly different from that of the control groups (*p* > 0.05). (Figure 2a) In order to prove the biocompatibility of IL-1ra CS/β-GP/gel hydrogel in vivo, we randomly divided rats into six groups for H&E staining of liver and kidney (Figure 2c), and there was no significant difference among the groups. In the analysis of liver and kidney indexes in rat serum, we found no significant difference in AST, ALT, and Cr (Figure 2d) in the rat serum of each group (*p* > 0.05).

### 2.3. IL-1ra CS/β-GP/Gel Hydrogel Decreased the mRNA Expression of Inflammatory Factors

Our in vitro (Figure 2b) and in vivo (Figure 3a) experiments showed similar mRNA expressions of inflammatory factors. The results of cell experiments showed that the mRNA expression of IL-1β, IL-6, and TNF-α of RAW264.7 cells stimulated with LPS was 1.58 times, 1.91 times, and 3.35 times higher than those of untreated cells, and the expression of these three cytokines increased by 2.16 times, 1.45 times, and 1.91 times, respectively, after high glucose culture. The expression of these inflammatory cytokines decreased significantly after treatment with IL-1ra CS/β-GP/Gel hydrogel extract. In animal experiments, the mRNA expression of IL-1β, IL-6, and TNF-α in rats with simple periodontitis was 2.63 times, 2.79 times, and 4.18 times higher than those in healthy rats, and the mRNA expression of these cytokines in rats with diabetes increased by 1.78 times, 1.58 times, and 3.04 times, respectively. The expression of inflammatory factors after treatment with IL-1ra alone and blank hydrogel was not significantly different from that in the diabetic periodontitis group but was significantly decreased in the IL-1ra CS/β-GP/Gel group (*p* < 0.05).

### 2.4. IL-1ra CS/β-GP/Gel Hydrogel Can Improve Blood Glucose Levels in Diabetic Rats with Periodontitis

To evaluate the effect of the hydrogel loaded with IL-1ra CS/β-GP/Gel on diabetic periodontitis rats, we monitored the changes in body weight (Figure 3b) and blood sugar of rats from the start of ligation. At this time, the rats injected with STZ had dull hair, listlessness, decreased activity, a slow response, significantly increased food and water consumption, increased urine output, and decreased body weight, which showed typical “three more and one less” symptoms, and the diabetes model was established. We observed that the weight of rats in the blank control group without any treatment showed a steady upward trend.

The rats in the ligation groups lost a little weight within 3 days after ligation, which may be due to maladaptation to the ligation intervention. After 3 days, the weight of the rats in the periodontitis group and hydrogel group loaded with IL-1ra CS/β-GP/Gel recovered and increased, while the weight of rats in the diabetic periodontitis group, IL-1ra treatment group, and blank hydrogel treatment group decreased gradually with time.

Our experimental results showed that the levels of blood glucose (32.80 ± 2.50 mmol/L) and glycosylated hemoglobin (13.30% ± 3.51%) in untreated diabetic periodontitis rats increased significantly, which agreed with the standard of diabetes in rats (Figure 3c). The levels of blood glucose and HbA1c in the IL-1ra group and the blank hydrogel group decreased, but there was no statistical difference between them and the diabetic periodontitis group. Serum glucose (9.80 ± 0.80 mmol/L, *p* < 0.05) and HbA1c (5.70% ± 0.69%, *p* < 0.0001) were significantly decreased in the IL-1ra CS/β-GP/Gel hydrogel group. The results of blood glucose level changes are shown in Table 1.

### 2.5. CT Evaluation of the Effect of IL-1ra CS/β-GP/Gel Hydrogel on Rat Alveolar Bone

The general oral view of rats showed that the gums of normal rats were pink and hard on palpation, while the gums of periodontitis rats were dark red and blood exudation was observed on exploration (Figure 4a). A large amount of food residue was observed in the mouth of diabetic periodontitis rats, covering all of the crowns of the first molars and 1/2 of the second molars in the right maxillary. In the rats treated with IL-1ra alone and blank hydrogel, the gingival bleeding was not significantly alleviated after exploration, and obvious food debris was still visible around the first molar. After treatment with IL-1ra CS/β-GP/Gel hydrogel, gingival bleeding was alleviated and no obvious food residue was observed.

The results of the Micro-CT scan (Figure 4b) showed that compared to the blank group and simple periodontitis group, the level of the alveolar bone in the diabetic periodontitis group decreased significantly, and the decreased alveolar bone was concave between the first and second molars. The mesiobuccal root, distal buccal root, and distal lingual root of the first molars were exposed, among which, the distal buccal lingual root was more than 2/3 exposed, and the mesiobuccal lingual root of the second molars was also partially exposed. The level of alveolar bone in the groups treated with IL-1ra and blank hydrogel increased, but it was far less than that in the group loaded with IL-1ra CS/β-GP/Gel hydrogel.

The ABL values measured by ImageJ were significantly different among the six groups (Figure 4c). The ABL values of the blank group (0.71 ± 0.12 mm) (*p* < 0.05) and periodontitis group (1.11 ± 0.22 mm) (*p* < 0.001) were significantly lower than those of the diabetic periodontitis group (1.64 ± 0.38). However, the difference in the IL-1ra CS/β-GP/Gel hydrogel treatment group was greater (0.96 ± 0.08 mm) (*p* < 0.0001), but there was no significant difference between the IL-1ra alone treatment groups (1.35 ± 0.13 mm) and diabetic periodontitis group. Further analysis of the region of interest between the first and second molars showed that the BV/TV in the diabetic periodontitis group decreased by 27.83% (*p* < 0.0001) and 22.70% (*p* < 0.001), respectively. Compared to the diabetic periodontitis group, the bone volume fraction of rats treated with IL-1ra alone and blank hydrogel increased by 14.95% (*p* < 0.05) and 20.85% (*p* < 0.01), but it was lower than that of the hydrogel group loaded with IL-1ra CS/β-GP/Gel (27.48%, *p* < 0.0001). Compared to the control group and simple periodontitis group, the Tb. Th in the diabetic periodontitis group decreased by 0.12 mm (*p* < 0.0001) and 0.05 mm (*p* < 0.001), respectively, and Tb. Th increased after treatment with IL-1ra CS/β-GP/Gel hydrogel (0.10 mm, *p* < 0.0001).

### 2.6. Anti-Inflammatory Properties of IL-1ra CS/β-GP/Gel Hydrogel In Vivo

We next stained the periodontal tissues of rats to further evaluate the experimental results of IL-1ra CS/β-GP/Gel hydrogel inhibiting periodontitis in diabetic rats. The results of H&E (Figure 5a) showed that collagen fibers were arranged orderly in the periodontal supporting tissues of healthy rats, the alveolar bone surface was smooth, no obvious osteoclasts and bone lacunae were formed, and the gingival epithelium became a slender strip with a complete structure. In contrast, the normal epithelial structure of periodontitis rats was destroyed, the combined epithelium extended to the root, and deep periodontal pockets were formed. Collagen fibers in the gingival sulcus epithelium and connective tissue below the combined epithelium presented with edema and degeneration, forming unstructured tissue, and extensive inflammatory cell infiltrate. Inflammation in rats with diabetes mellitus was more obvious, and the alveolar bone was absorbed and destroyed to different degrees. After IL-1ra CS/β-GP/Gel hydrogel treatment, the gingival epithelial structure was destroyed, but the inflammatory cell infiltration decreased. There was no significant difference in staining results between the IL-1ra alone and blank hydrogel treatment groups and diabetic periodontitis group.

Immunohistochemical staining of rat periodontal tissue sections with IL-1β and IL-6 antibody (Figure 5b) showed that, compared to the control group, there were obvious positive areas in the periodontitis group, with proportions of IL-1β-positive cells up to 9.06% ± 5.42% and IL-6 up to 20.29% ± 4.97%. The results showed that ligation was effective in establishing the periodontitis model in rats. The proportion of Il-1β (35.11% ± 4.53%, *p* < 0.0001) and IL-6 (41.02% ± 10.51%, *p* < 0.0001) positive areas and positive cells in untreated diabetic rats with periodontitis were significantly higher than those in untreated diabetic rats with periodontitis. These results indicate that diabetes can further aggravate the occurrence and development of periodontal inflammation. Il-1β (23.43% ± 8.01%) and IL-6 (30.43% ± 7.39%) positive cells were partially reduced in the positive area of the IL-1ra group compared to the periodontitis group of diabetic rats, but the proportion of positive cells was still much higher than that of the blank control and periodontitis groups. There was no significant difference in the proportion of IL-1β (27.07% ± 14.23%) and IL-6 (34.49% ± 12.04%) positive cells between the blank hydrogel group and the diabetic periodontitis group, indicating that IL-1ra alone and hydrogel could not effectively inhibit periodontal inflammation. Positive regions still existed in the IL-1ra CS/β-GP/Gel hydrogel group but were significantly smaller than those in the diabetic periodontitis group. The proportion of IL-1β (11.91% ± 6.10%) and IL-6 (11.81% ± 2.94%) positive cells was statistically significant, and the IRS decreased from 6 to 1.2 after treatment. The results of immunohistochemistry are summarized in Table 2.

## 3. Discussion

At present, oral hygiene instruction, scaling and subgingival curettage, and scaling and root planing (SRP) are the most basic and effective treatments for periodontal disease. However, affected by dental anatomical conditions such as root-forking lesions and periodontal pocket depth, SRP alone often cannot eliminate plaque microorganisms [14]. Therefore, local drug therapy has become an important adjunctive therapy for chronic periodontitis. Additionally, for patients with periodontitis with type 2 diabetes, poor blood glucose control will increase periodontal tissue damage and aggravate the development of periodontal disease; this often leads to periodontal inflammation and periodontal tissue trauma caused by prolonged, poor clinical treatment [3,4,5]. CS/β-GP/Gel hydrogel is an injectable temperature-sensitive hydrogel, which has been widely used in the field of tissue engineering as the carrier of seed cells and biological factors. Some scholars have used the chitosan/β-glycerophosphate/collagen (CS/β-GP/Co) hydrogel of tendon-carrying stem cells (TSCs) to promote the healing of acute Achilles tendon injury in rats [15]. Moreover, injecting CS/β-GP/Gel hydrogel loaded with collagenase into the tendon-bone interface of rabbits can promote the early healing of the tendon-bone interface and the formation of new bone [16]. In this experiment, three types of raw materials were mixed and formed into a gel at 37 °C, with good morphology and uniform pore size distribution, demonstrating that the hydrogel could realize the sustained release of drugs, and was beneficial to the entry and exit of nutrients and metabolic waste. Comparing the infrared spectra of CS/β-GP/Gel hydrogel and its components, the absorption peak of CS at 3113.4 cm^−1^ moves to 2919.9 cm^−1^ in the CS/β-GP/Gel at low frequency, which indicates a coordination bond between chitosan and β-GP. Moreover, the formation of a coordination bond changes the electron cloud of the CS amino N, which leads to the weakening of the N-H bond and subsequent stretching vibration and bending vibration. The absorption peaks at 1588.9 cm^−1^ and 1498.0 cm^−1^ in chitosan hydrogel show that covalent and hydrogen bonds are formed between amino and hydroxyl groups on chitosan and gelatin molecules. XRD showed that the diffraction peaks of the CS and gel were weakened due to the action of -OH and PO43- after β-GP was added, which destroyed the crystalline state of CS. The addition of gelatin led to the formation of a strong hydrogen bond between gelatin and chitosan, weakened the hydrogen bond between -NH2 and -OH in chitosan molecules, and destroyed the ordered structure of chitosan, thus reducing the crystallinity of chitosan and gelatin. The above characterization indicated that CS/β-GP/Gel thermosensitive hydrogel was successfully cross-linked by CS, β-BP, and Gel. Moreover, the in vitro and in vivo studies showed that the hydrogel loaded with IL-1ra CS/β-GP/Gel has good biocompatibility and is a suitable drug carrier.

Although chitosan, as one of the raw materials of CS/β-GP/Gel hydrogels, has hypoglycemic properties, the chitosan used for hydrogels has a large molecular weight (more than 1 million) and poor water solubility, which greatly limits its hypoglycemic effects [17,18,19,20,21]. STZ is a cytotoxic glucose analog, which is absorbed by islet β cells through the GLUT2 glucose transporter. After being absorbed into cells, STZ inhibits DNA synthesis by inducing DNA division and methylation, resulting in cell death [22]. Recent studies have shown that mesenchymal stem cells can reverse the dedifferentiation of islet β cells through IL-1ra, thus alleviating the dysfunction of islet β cells [23,24]. Clinical trials have confirmed that IL-1ra can increase insulin secretion by 2.5 times in patients with newly diagnosed type I diabetes [23]; however, the half-life of IL-1ra is short, only 4–6 h, and the ideal hypoglycemic effect can only be achieved by repeated daily administration at a dosage far higher than our experimental dosage [10]. Therefore, IL-1ra alone could not effectively reduce blood sugar. Our CS/β-GP/Gel hydrogel maintains the drug concentration in the local periodontal pocket within the effective concentration range for a longer duration. The controlled-release administration mode is ideal to effectively improve the hyperglycemia level of diabetic rats, which is consistent with our experimental results. After treatment with IL-1ra CS/β-GP/Gel hydrogel, the weight loss of diabetic periodontitis rats decreased, possibly due to an improvement in the severity of diabetes promoting weight gain. Furthermore, the decrease in blood glucose level and glycosylated hemoglobin proved that IL-1ra CS/β-GP/Gel hydrogel could improve the hyperglycemia level in diabetic rats.

Periodontitis is a multifactorial disease in which dental plaque is the initial factor. Plaque can induce early inflammatory processes, while local factors, such as poor oral hygiene and occlusal trauma, and systemic factors, such as endocrine disorders and genetic factors, affect the occurrence and development of periodontitis [2,25]. The components of dental plaque biofilm can stimulate host cells to produce proinflammatory cytokines, such as IL-1β and TNF-α [26,27,28]. IL-1β is a potential stimulator of osteoclast proliferation, differentiation, and activation, which can induce interstitial cells to produce proteases, including metalloproteinases (MMPs), which are involved in connective tissue destruction [29,30,31,32]. TNF-α also mediates a series of biological processes and can induce the destruction of connective tissue and alveolar bone [33]. These biological processes include stimulating bone resorption, inhibiting bone formation, inhibiting proteoglycan synthesis, inducing collagen and cartilage to degrade metalloproteinase and prostaglandin E2, and further producing TNF and other pro-inflammatory cytokines. IL-6 also plays a key role in the initial and acute stages of periodontitis [34]. In addition to its role in the immune response, IL-6 participates in alveolar bone homeostasis by increasing the expression of the RANKL receptor activator in osteoblasts, thus further promoting osteoclast differentiation and bone resorption. Researchers initially found that injecting IL-1ra into the knee joint of patients with rheumatoid arthritis can relieve pain [35]. Later, Nixon et al. [36] used recombinant IL-1ra for local administration and achieved obvious therapeutic effects in mice and Malaysian arthritis animal models. Moreover, HDAd-IL-1ra inhibited the inflammatory effects of factors such as IL-1β, prevented the development of cartilage injury and synovitis, and achieved good anti-inflammatory effects. Deborah J Gorth et al. [37] prepared IL-1ra/PLGA microspheres using the double emulsion solvent evaporation method. The results showed that IL-1ra was slowly released after 24 h by a sudden release effect and 20 mg had accumulated on the 35th day. In addition, the mRNA expression of iNOS, ADAMTS-4, MMP-13, IL-1β, IL-6, and toll-like receptor 4 (TLR-4) were significantly inhibited by co-culture with mature nucleus pulposus cells. Moreover, Bo Qiu et al. [38] successfully prepared IL-1ra hyaluronic acid chitosan microspheres, with no obvious sudden release effect, which released linearly for up to 8 days. ELISA results showed that HA-CS-IL-1ra microspheres effectively inhibited the biological activity of IL-1β by inhibiting the secretion of NO_2_^−^ and PGE2. Our previous experiments also proved that IL-1ra-loaded sustained-release microspheres can play a role in the treatment of periodontitis [39]. However, many studies have shown that the microsphere sustained-release preparation has a sudden release effect, despite the long sustained-release time. The severe mechanical action and physical and chemical conditions, such as the organic solvents in the preparation process, may lead to the denaturation of protein drugs. However, the temperature-sensitive hydrogel loaded with IL-1ra CS/β-GP/Gel used in this experiment can release IL-1ra continuously for up to 21 days, ensuring the sustained inhibition of inflammation. Moreover, PCR showed that the expression of IL-1β, IL-6, and TNF-α in cells decreased significantly after treatment with IL-1ra hydrogel.

Previous studies have shown that ligating the neck of rat teeth with silk thread or ligating wire as a local stimulating factor can lead to the formation of histopathological manifestations similar to those of human periodontitis to varying degrees [40,41]. Local wire ligation can cause the accumulation of plaque, soft scale, and tartar, and form a local stimulating environment. Local ligation is simple, and early bone resorption is obvious. However, new bone formation may occur over time, while bone loss may decrease, making the model more suitable for short-term study. Two weeks after ligation, compared to the control group, the rats showed redness and swelling of the gums, and infiltration of neutrophils in the epithelium and connective tissue, indicative of the acute stage of periodontitis. The symptoms were worse at 4 weeks after ligation than at 2 weeks, and the gums showed a severe inflammatory reaction. The redness and swelling of gingival epithelium showed severe erosion and even ulcer formation, and a large number of lymphocytes infiltrated in subepithelial connective tissue; collagen degeneration and destruction and alveolar bone destruction were obvious [42]. Therefore, we removed the ligature wire 4 weeks after ligating the maxillary first molars of rats to observe the inflammatory reaction in the periodontal tissues and observed the changes in the alveolar bone at 4 weeks. The results of tissue PCR were consistent with those of cells. The expression of IL-1β, IL-6, and TNF-α in the periodontal tissues of the IL-1ra hydrogel treatment group decreased significantly. Moreover, IHC showed that the positive areas of IL-1β, IL-6, and TNF-α in the IL-1ra hydrogel treatment group decreased, as did the IRS, demonstrating the effective inhibition of inflammatory factors in periodontal tissues of diabetic rats. The results of HE and Micro-CT showed that the IL-1ra hydrogel could reduce inflammatory cell infiltration in periodontal tissue and relieve alveolar bone absorption, demonstrating great potential for treating periodontitis.

There are still some limitations in this study. Micro-CT results showed that hydrogel reduces the absorption of alveolar bone. Therefore, we can further explore whether the hydrogel has osteogenic properties by changing the levels of osteoblasts in vivo and in vitro, and further study the potential pathways of anti-inflammation and osteogenesis on this basis, which will further improve this experiment. In addition, previous studies have shown that effective treatment of periodontal inflammation can promote a decrease in glycosylated hemoglobin levels [5,43]. Therefore, in our experimental results, the decrease in blood glucose level in rats may also be related to the fact that IL-1ra CS/β-GP/Gel thermosensitive hydrogel can effectively relieve local periodontal inflammation and then affect the whole blood glucose level. However, the reason why IL-1ra-loaded thermosensitive hydrogel lowers blood sugar is still unclear. It may be due to the direct effect of IL-1ra on diabetes, or the indirect reduction in blood sugar due to the reduction in local periodontitis after hydrogel treatment, or that both of the above factors exist at the same time. The above-mentioned related mechanism can also be taken as the research focus. Therefore, this experiment still has great research potential to be further explored.

## 4. Materials and Methods

### 4.1. Synthesis of IL-1ra CS/β-GP/Gel Temperature-Sensitive Hydrogel

To synthesize IL-1ra CS/β-GP/Gel temperature-sensitive hydrogel, 1 g CS (molecular weight: 310,000–375,000 Da, Sigma-Aldrich, Munich, Germany) was weighed onto weighing paper (10 cm × 10 cm, Hangzhou Fuyang Chengkun Experimental Instrument Co., Ltd., Hangzhou, China), and irradiated under an ultraviolet lamp for 1 h (30 min per side). Following UV sterilization, the CS was dissolved in 50 mL 0.1 mol/L HCl aqueous solution to prepare a 2% (*w*/*v*) CS solution. Next, 5.6 g β-GP was dissolved in 10 mL deionized water and configured with a mass fraction of 56% β-GP solution (including 50 μg/mL IL-1ra). Subsequently, 0.2 g gelatin (molecular weight: 40,000–50,000 Da, Sigma-Aldrich, Munich, Germany) was dissolved in 40 mL deionized water to prepare an aqueous gelatin solution with a mass fraction of 0.5%. The β-GP solution and gelatin solution were filter sterilized using a 0.22 μm disposable filter before use. The CS solution, β-GP solution, and gelatin solution were configured in a 40:8:1 volume ratio. Next, 20 mL CS solution was transferred to a beaker and magnetically stirred for 10 min, before adding 4 mL β-GP solution dropwise in an ice bath. When the solution became turbid, 500 μL gelatin aqueous solution was added and magnetically stirred until the solutions were fully mixed. The pH was adjusted to 7.2 with 0.1 mol/L NaOH to obtain CS/β-GP/Gel temperature sensitive hydrogel loaded with IL-1ra. Next, 2 mL hydrogel was transferred into a sterile ampoule, which was sealed and placed in a constant temperature water bath at 37 °C. The gel time was recorded by the tube inversion method.

### 4.2. Characterization of CS/β-GP/Gel Thermosensitive Hydrogels

The IL-1ra CS/β-GP/Gel temperature-sensitive hydrogel was pre-frozen at –80 °C for 24 h and then freeze-dried. The microstructure of the temperature-sensitive hydrogel was observed by scanning electron microscopy (SEM; SUPRA 55 SAPPHIRE, Carl Zeiss Jena, Germany). Next, 1 mg of freeze-dried IL-1ra CS/β-GP/Gel-loaded temperature-sensitive hydrogel was mixed with an appropriate amount of Kbr powder and tableted. The structure and composition of the organic functional groups of the temperature-sensitive hydrogel were analyzed by Fourier transform infrared spectrometry (FTIR, Nicolet 560 FTIR spectrometer, Nicolet Instrument Corporation, Madison, WI, USA) in the range of 400–4000 cm^−1^. The freeze-dried hydrogel samples were crushed with a mortar, and the phase structure of the hydrogel was analyzed by an X-ray diffractometer (Bruker AXS GmbH, Karlsruhe, Germany), with the following working conditions: Cu target Kα radiation; Ni as a filter; tube pressure, 60 kV; working current, 40 mA; scanning range, 5–40°; scanning speed, 6 (°)/min; and resolution, 0.02%.

In order to measure the degradation rate and drug release rate of CS/β-GP/Gel thermosensitive hydrogel in vitro, we accurately measured 2 mL of IL-1ra CS/β-GP/Gel thermosensitive hydrogel with a pipette at 4 °C, put it in a 5 mL centrifuge tube, and incubated in a constant temperature incubator at 37 °C for 3 h to ensure that it can be completely transformed into gel. Then, add 2 mL of SBF into the centrifuge tube, and keep incubating in the incubator at 37 °C, and discard the buffer every seven days. The following is the calculation formula of the degradation rate $(t)\%=\frac{w0-wt}{w0}\times 100\%$. W_0_ is the initial gel dry weight and W*_t_* is the dry weight of gel at the moment of degradation.

In the aseptic environment, the mixed solution of IL-1ra CS/β-GP/Gel before gelation was added to a 6-well plate (2 mL per well), and the plate was incubated in an incubator at 37 °C to fully gel. Following incubation, 8 mL SBF was added to each well and further incubated at 37 °C. Following incubation, 3 mL supernatant was collected from each well at 1 h, 2 h, 3 h, 8 h, 24 h, 48 h, 72 h, and 5, 7, 10, 14, and 21 days after incubation, and the wells were supplemented with an equal volume of SBF solution. According to the standard curve in Enzyme-linked immunosorbent assay (ELISA; Dakewe, Beijing, China), the concentration of IL-1ra in supernatant at each time point was quantitatively analyzed. Then calculate the cumulative release rate according to the following formula: $Fi=(3\sum C_{i-1}+8Ci)/(\mathrm{dosage}\;\times \mathrm{drug}\;\mathrm{content})$. F*_i_* is the cumulative release rate of the system after the *i*th sampling, C*_i_* is the drug release concentration at the *i*th sampling, 3 is the volume of each sampling (unit: mL), and 8 is the total volume of the release system (unit: mL), thus drawing the cumulative release curve.

### 4.3. Cytocompatibility of IL-1ra CS/β-GP/Gel Temperature-Sensitive Hydrogels

In the aseptic environment, the mixed solution of CS/β-GP/Gel was added to a 6-well plate in a volume of 900 μL per well, and then allowed to fully gel in an incubator at 37 °C. After gelling, 3 mL high-sugar medium was added to each well, and the incubation was continued for 24 h in an incubator at 37 °C to prepare an extract. Following incubation, the extract was filtered with a disposable 0.22 μm filter, and fetal bovine serum (FBS) and penicillin-streptomycin solution were added to prepare the extract. Then, mouse macrophages (RAW264.7, Thermo Fisher Scientific, Waltham, MA, USA), which grow well under incubation conditions of 37 °C, 5% CO2, and 100% humidity, were evenly mixed with the extract, and then inoculated into two 96-well plates at a density of 2 × 10^3^/well and incubated for 24 h until adherent. After 24 h, the original medium was replaced with extracts containing different concentrations of IL-1ra (1, 10, 20, 50, 100, 150, 200 μg/mL, *w*/*v*) for 24 h and 48 h. Next, 10 μL of CCK-8 reagent was added to each well, before reacting at 37°C for 2 h. The absorbance (OD) at 450 nm was detected using a microplate reader (RT-6000; Lei Du Life Science and Technology Co, Shenzhen, China), and the cell viability was calculated as follows: Cell survival rate = [(experimental well OD value − blank well OD value)/(control well OD value − blank well OD value)] × 100%. The experiment was repeated thrice.

### 4.4. IL-1ra CS/β-GP/Gel Thermosensitive Hydrogel Inhibits Inflammatory Cytokine Expression in Macrophages under High Glucose Conditions In Vitro

Lipopolysaccharide is the main component of the cell wall of Gram-negative bacteria and can induce macrophages to produce various pro-inflammatory factors, including IL-1β, IL-6, and TNF-α. During periodontal infection, LPS and various inflammatory factors can inhibit osteoblast activity, promote osteoclast activation and proliferation, and break the periodontal steady state, all of which can indirectly or directly lead to alveolar bone absorption [44]. Therefore, LPS stimulation of RAW264.7 cells with Porphyromonas gingivalis was used to establish the inflammation model in vitro, with high glucose stimulation given simultaneously (50 mmol glucose per elevated glucose medium); 5.5 mmol/L low-glucose medium was used as the base medium, and 406 mg glucose powder was dissolved in 30 mL DMEM until fully dissolved. Next, the solution was filter sterilized with a 0.22 μm filter before adding DMEM, FBS, and double antibody to a constant volume of 50 mL. To evaluate the anti-inflammatory effect of IL-1ra CS/β-GP/Gel temperature-sensitive hydrogels, we measured the mRNA expression of inflammatory cytokines in RAW264.7 cells under high glucose conditions in vitro by quantitative real-time polymerase chain reaction. RAW264.7 cells were cultured under the conditions outlined in Section 2.5. The specific groups were as follows: (1) Blank control group: RAW264.7 without any treatment; (2) LPS stimulation group: RAW264.7 cells were stimulated with LPS (5 μg/mL) for 24 h; (3) LPS + high glucose stimulation group: RAW264.7 cells were stimulated with LPS (5 μg/mL) in high glucose medium (containing 50 mmol/L glucose) for 24 h; (4) LPS + high glucose stimulation + blank hydrogel group: high glucose medium (containing 50 mmol/L glucose) and blank hydrogel (CS/β-GP/Gel temperature-sensitive hydrogel without IL-1ra) were co-cultured for 24 h. The collected high glucose gel extract was used to stimulate RAW264.7 cells for 3 h, while LPS (5 μg/mL) was used to stimulate RAW264.7 cells for 24 h; and (5) LPS + high glucose stimulation + IL-1ra CS/β-GP/Gel thermosensitive hydrogel group: CS/β-GP/Gel temperature-sensitive hydrogel loaded with IL-1ra was cultured in high glucose medium (containing 50 mmoL/L glucose) for 24 h. The collected high glucose hydrogel extracts were used to stimulate RAW264.7 cells for 3 h, and LPS (5 μg/mL) was used to stimulate RAW264.7 cells for 4 h. According to Lu’s method [39], RT-qPCR was used to detect the mRNA expression of IL-1β, IL-6, and TNF-α in each group. The primer sequences used in RT-qPCR are shown in Table 3.

### 4.5. In Vivo Toxicity of IL-1ra CS/β-GP/Gel Thermosensitive Hydrogel

We used male Wistar rats, 6–8 weeks old, average weight 240–260 g. Thirty-six male Wistar rats were randomly divided into six groups: blank group (group I), periodontitis group (group II), diabetic periodontitis group (group III), diabetic periodontitis + blank hydrogel treatment group (group IV), diabetic periodontitis + IL-1ra alone group (group V), and diabetic periodontitis + IL-1ra CS/β-GP/Gel thermosensitive hydrogel treatment group (group VI). The rat model of diabetes was established by intraperitoneal injection of streptozotocin (STZ) 60 mg/kg. On the 3rd and 7th day after injection, blood was collected from the tail vein to detect the blood glucose level of rats; a blood glucose level > 16.7 mmol/L recorded twice indicated that the modeling was successful. The model of diabetes mellitus was successfully established, and the rat periodontitis model was established simultaneously; that is, the rats were anesthetized by intraperitoneal injection of 1% pentobarbital sodium. A 0.2 mm orthodontic ligature wire was placed between the first molar and the second molar of the maxilla of the rats. On the day when the periodontitis model was established, 50 μL of thermosensitive hydrogel containing 50 μg/mL IL-1ra was injected into the buccal-palatal gingival mucosa of the maxillary first molar of the rat model side. Inject once a week until the end of the experiment, a total of four times. Rats were anesthetized by intraperitoneal injection of 1% pentobarbital sodium after 4 weeks, and then euthanized. The bilateral maxilla, periodontal tissues, and liver and kidney tissue of rats were taken for subsequent experiments.

Four weeks after modeling, the rats were anesthetized with 1% pentobarbital sodium intraperitoneally. Next, 3 mL blood was collected from the right ventricle and supernatant was taken by a static method to detect the levels of serum creatinine (Cr), alanine aminotransferase (ALT), and aspartate aminotransferase (AST). After fixation for 48 h, the liver and kidney tissue samples were rinsed with water to remove the tissue fixative fluid, dehydrated and embedded, and then sectioned for hematoxylin and eosin (H&E) staining for histological analysis.

### 4.6. Effect of IL-1ra-Loaded CS/β-GP/Gelatin Thermosensitive Hydrogel on Body Weight and Blood Glucose in Rats and Its Anti-Inflammatory Properties In Vivo

The weight of rats was recorded from the first day of ligation. Three rats in each group were randomly selected and weighed at 12 noon every day, and the average value was taken for two weeks. After the establishment of the diabetes model, tail venous blood of rats in each group was collected at the same time every week, and the blood glucose level was monitored by a glucose meter at the end of the experiment. The collected serum was thawed again, and the serum glycosylated hemoglobin (HbA1c) level of rats in each group was determined by ELISA (ELISA; Meimian, Suzhou, Jiangsu, China).

The total RNA of rat maxillary alveolar bone preserved in liquid nitrogen was extracted by TRIzol (Life Technologies, Carlsbad, CA, USA), and then reverse-transcribed and amplified by qRT-PCR (Takara, Kusatsu, Shiga, Japan). The grouping was as outlined in Section 4.5, and the statistical analysis method was the same as that outlined in Section 4.4.

### 4.7. Micro-CT

The alveolar bone of rats was fixed with 4% paraformaldehyde for 48 h, and then fixed in a scanning container for CT scanning (SkyScan 1172; Bruker, Germany, 24 kV, 2 mA, 90 s). The level of alveolar bone between the first and the second molars was analyzed in the sagittal plane. The three-dimensional images were reconstructed by IPL image processing software (Scanco, Bassersdorf, Zurich, Switzerland). The distance from the cementoenamel junction (CEJ) to the alveolar bone crest (ABC) of each sample was taken as an ABL value (ABL = CEJ − ABC). For each tooth, we recorded six points in the buccal mesial, central and distal, and palatal mesial, central, and distal. The alveolar bone between the first and second molars was analyzed using bone parameter analysis software, including the bone volume fraction (BV/TV) and trabecular thickness (Tb.Th).

### 4.8. Histological Evaluation of Periodontitis in Diabetic Rats with IL-1ra CS/β-GP/Gel Thermosensitive Hydrogel

The Micro-CT treated samples were submerged in 10% EDTA for decalcification for 3 months, and the decalcified solution was changed every 2 days. After decalcification, the tissues were dehydrated, embedded, and sliced. The slicing was performed along the long axis of the first premolars, and the slices were evenly cut from the middle to the center and the far middle, with a thickness of approximately 3 μm. H&E staining was performed to evaluate the inflammation of rat periodontal tissue. IL-1β and IL-6 antibodies (Sangon, Shanghai, China) were used to perform immunohistochemical staining (IHC) on the tissue sections, and the staining of the upper and the first molar and the second molar (the ligation site) was observed with an optical microscope (BX51, Olympus, Tokyo, Japan). At least five fields of view with a magnification of ×400 were randomly selected, and the immune response score (IRS) of the area was calculated. The presence of a brownish yellow precipitate was regarded as an immunohistochemical positive reaction, and the cell staining intensity (SI) was divided into four grades as follows: no positive staining or negative score, 0 point; light yellow or weak positive score, 1 point; brown or positive score, 2 points; and brown or strong positive score, 3 points. The percentage of positive cells (PP) was divided into four grades as follows: the number of positive cells accounts for ≤25% of the total number of cells, score 1 point; 26–50%, 2 points; 51–75%, 3 points; and >75%, 4 points. The IRS was calculated as follows: IRS = SI × PP.

### 4.9. Statistical Analysis

All of the in vivo and in vitro experiments were performed in triplicate and repeated thrice. The experimental data were statistically analyzed using SPSS v23.0 software and plotted with Prism GraphPad v8.0 software (GraphPad by Dotmatics, San Diego, CA, USA). One-way ANOVA was used to compare the difference among three groups, and all calculated data are expressed as $\bar{x}$ ± s. *p*-values < 0.05 were considered to be statistically significant.

## 5. Conclusions

IL-1ra CS/β-GP/Gel thermosensitive hydrogel could realize long-term stable drug release, effectively inhibit the expression of inflammatory factors, partially alleviate the hyperglycemia level, continuously and effectively produce anti-inflammatory effects in periodontal tissues, and significantly inhibit alveolar bone loss in a diabetic rat model. Therefore, in situ injection of a IL-1ra thermosensitive hydrogel scaffold is an effective and innovative drug and treatment for treating patients with chronic periodontitis with diabetes.

## Figures and Tables

**Figure 1 ijms-23-13939-f001:**
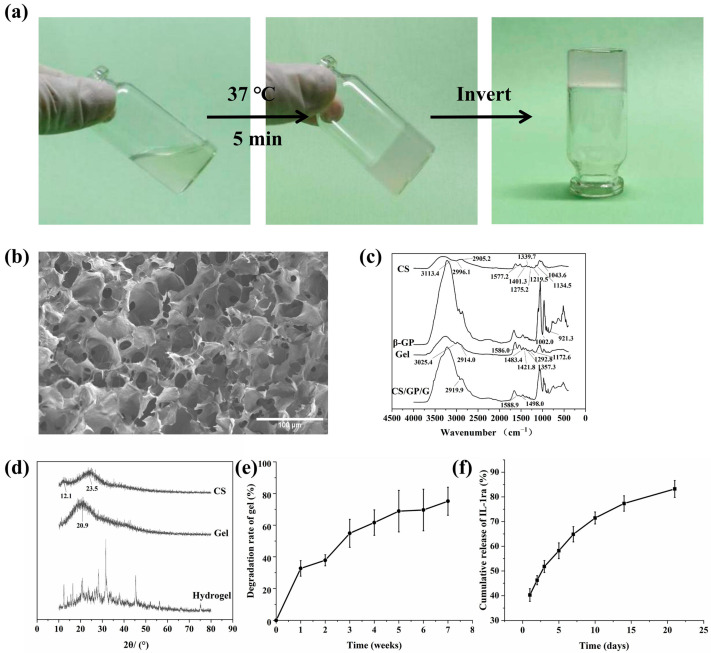
Characterizations of CS/β-GP/gelatin thermosensitive hydrogels. (**a**) Image of CS/β-GP/gelatin mixed solution transformed into hydrogel after 5 min at 37 °C. (**b**) Optical microscope images of chitosan/β-GP/gelatin thermosensitive hydrogel. (**c**) FT-IR spectra for chitosan, β-GP, gelatin, and chitosan/β-GP/gelatin thermosensitive hydrogel. (**d**) XRD patterns of chitosan, gelatin, and chitosan/β-GP/gelatin thermosensitive hydrogel. (**e**) Degradation percentage curve of CS/β-GP/gelatin hydrogel. (**f**) The release percentage curve of IL-1ra from CS/β-GP/gelatin hydrogel.

**Figure 2 ijms-23-13939-f002:**
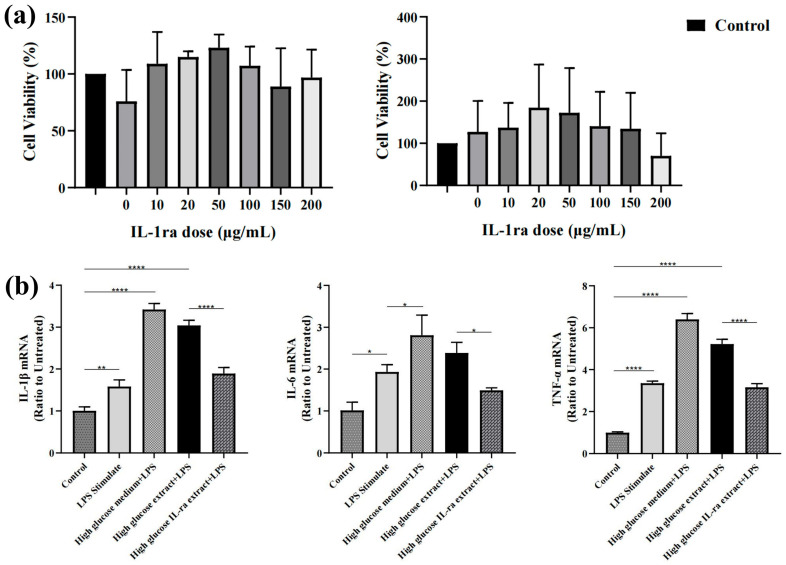
Biocompatibility and anti-inflammatory factor characteristics of IL-1ra-loaded CS/β-GP/gelatin thermosensitive hydrogel. (**a**) Viability of RAW 264.7 cells cultured with IL-1ra-loaded CS/β-GP/gelatin thermosensitive hydrogels at different concentrations (10–200 μg/mL) at 24 h and 48 h. (**b**) Fold change of IL-1β, IL-6, and TNF-α mRNA expression in the blank control, LPS stimulated, High glucose medium+ LPS stimulated, High glucose extract + LPS stimulated, High glucose extract + LPS + IL-1ra-loaded CS/β-GP/gelatin thermosensitive hydrogel groups (*n* = 3). (**c**) H&E staining of liver and kidney tissue in the groups I, II, III, IV, V, and VI, scale bar represents 100 μm. (**d**) Effect of IL-1ra-loaded CS/β-GP/gelatin thermosensitive hydrogels on serum creatinine (Cr), aspartate transaminase (AST), and alanine transaminase (ALT) levels. *: *p* < 0.05, **: *p* < 0.01, ****: *p* < 0.0001.

**Figure 3 ijms-23-13939-f003:**
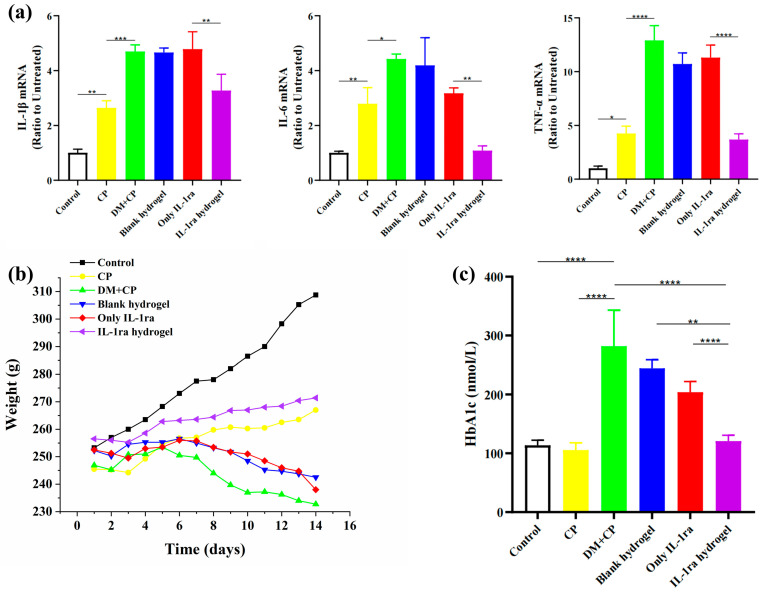
Anti-inflammatory effect of IL-1ra-loaded CS/β-GP/gelatin thermosensitive hydrogel in vivo and its effect on body weight and blood sugar of rats. (**a**) Fold change of IL-1β, IL-6, and TNF-α mRNA expression in groups I, II, III, IV, V, and VI. (**b**) Effect of IL-1ra-loaded CS/β-GP/gelatin thermosensitive hydrogel administration on rat weight. A black line is used to indicate the 14-day weight change in the I group to make the image results more intuitive. (**c**) The content of glycosylated hemoglobin in groups I, II, III, IV, V, and VI. *: *p* < 0.05, **: *p* < 0.01, ***: *p* < 0.001, ****: *p* < 0.0001.

**Figure 4 ijms-23-13939-f004:**
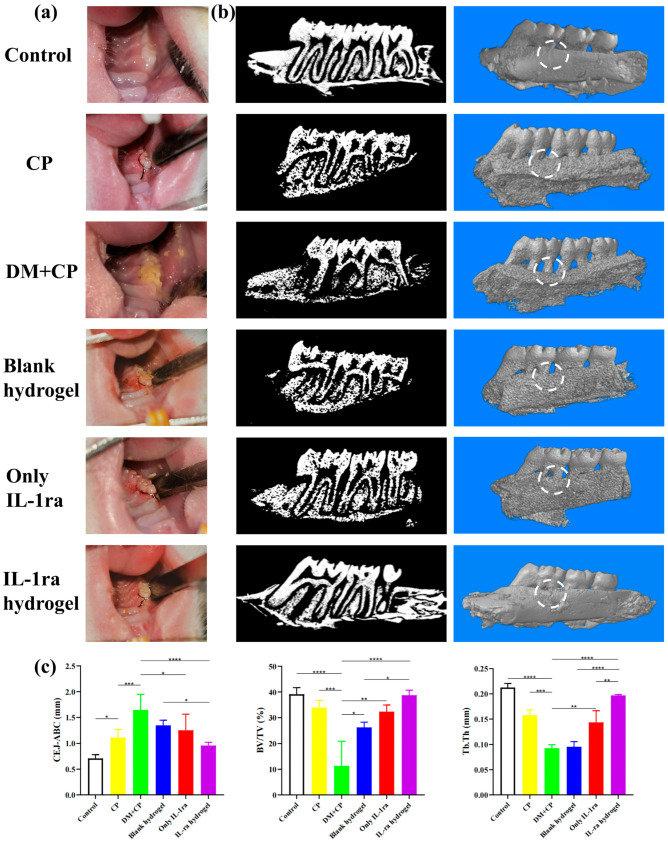
Effect of IL-1ra-loaded CS/β-GP/gelatin thermosensitive hydrogel on alveolar bone in rats. (**a**) Photos of periodontitis models induced by ligature around maxillary first molars of rats in groups I, II, III, IV, V, and VI. (**b**) Micro-CT images of maxillary alveolar bone surrounding the maxillary first molars and maxillary second molars four weeks after the treatment: the images with black background are representative sagittal Micro-CT slices; the images with blue background are three-dimensional Micro-CT reconstruction images. The circled area is the subsequent analysis area of alveolar bone parameters. (**c**) Quantitative analysis of ABL, BV/TV, and Tb. Th determined by Micro-CT images. *: *p* < 0.05, **: *p* < 0.01, ***: *p* < 0.001, ****: *p* < 0.0001.

**Figure 5 ijms-23-13939-f005:**
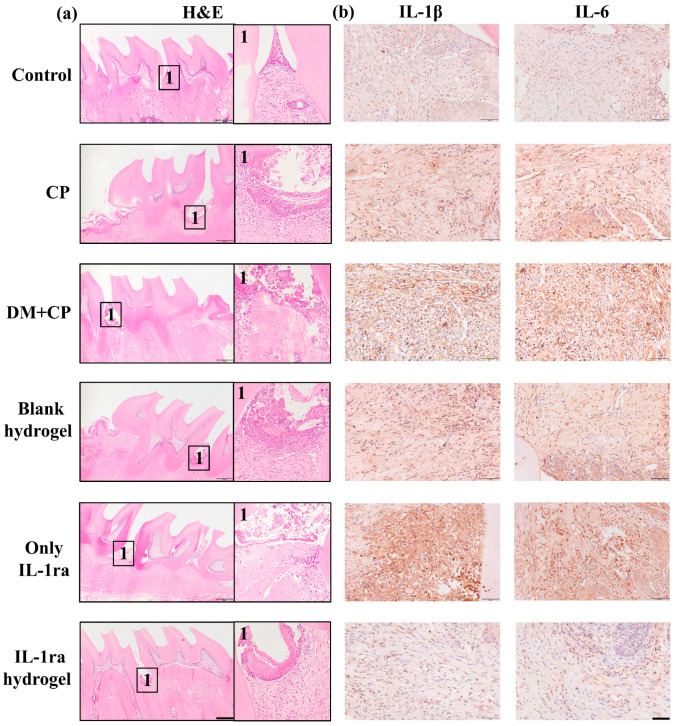
Immunohistochemical analysis of periodontal tissue sections in rats. (**a**) H&E image of the periodontal tissue section. The marked area is the periodontal tissue between the first molar and the second molar of rat maxilla, the part marked 1 in the figure shows more clearly and intuitively the expression of alveolar bone and gingival epithelium between the first molar and the second molar, scale bar represents 500 μm. (**b**) The results of IHC staining of IL-1β and IL-6 in periodontal tissues of rats in each group after 4 weeks of implantation, scale bar represents 50 μm.

**Table 1 ijms-23-13939-t001:** Characteristics of rats in groups I, II, III, IV, V, and VI.

Group	Bodyweight (g)	Blood Glucose (mmol/L)	HbA1c (%)
Control	308.75 ± 14.88 ****	7.70 ± 0.40 **	5.36 ± 0.64 ****
CP	267.00 ± 17.50 **	8.30 ± 0.30 **	5.25 ± 1.08 ****
DM + CP	232.75 ± 6.75	32.80 ± 2.50	13.30 ± 3.51
DM + CP + blank hydrogel	242.50 ± 13.50	28.60 ± 1.70	11.54 ± 0.81
DM + CP + IL-1ra	238.00 ± 16.00	26.10 ± 2.20	9.62 ± 1.10
DM + CP + IL-1ra loaded hydrogel	271.40 ± 12.00 ***	9.80 ± 0.80 *	5.70 ± 0.69 ****

*: *p* < 0.05, **: *p* < 0.01, ***: *p* < 0.001, ****: *p* < 0.0001 compared with untreated diabetic periodontitis rats.

**Table 2 ijms-23-13939-t002:** The proportion of different protein positive cells in diabetic periodontitis rat models.

Group	Proteins			
	IL-1β		IL-6	
	PP	IRS	PP	IRS
Control	0.22 ± 0.41 ****	0.40	0.17 ± 0.69 ****	0.20
CP	9.06 ± 5.42 ****	1.00	20.29 ± 5.97 ****	2.20
DM + CP	35.11 ± 4.53	6.00	41.02 ± 10.51	5.00
DM + CP + blank hydrogel	27.07 ± 14.23	4.00	34.49 ± 12.04	4.40
DM + CP + IL-1ra	23.43 ± 8.01 *	4.00	30.43 ± 7.39 *	5.20
DM + CP + IL-1ra loaded hydrogel	11.91 ± 6.10 ****	1.20	11.81 ± 2.94 ****	1.20

*: *p* < 0.05, ****: *p* < 0.0001 compared with untreated diabetic periodontitis rats.

**Table 3 ijms-23-13939-t003:** Primer sequences used for qRT-PCR.

Gene	Gene Bank	Sequence of Primers (5′-3′ Direction)	Length (bp)	Product (bp)
β-Actin	NM_031144.3	F: GGAGATTACTGCCCTGGCTCCTA	23	150
		R: GACTCATCGTACTCCTGCTTGCTG	24	
IL-6	NM_012589.2	F: AAGCCAGAGCTGCAGGATGAGTA	23	150
		R: TGTCCTGCAGCCACTGGTTC	20	
TNF-α	NM_012675.3	F: TTCCAATGGGCTTTCGGAAC	20	118
		R: AGACATCTTCAGCAGCCTTGTGAG	24	
IL-1β	NM_031512.2	F: CCCTGAACTCAACTGTGAAATAGCA	25	111
		R: CCCAAGTCAAGGGCTTGGAA	20	

## Data Availability

The data presented in this study are available on request from the corresponding authors.

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
