# Peer review of "Treatment of Periodontal Inflammation in Diabetic Rats with IL-1ra Thermosensitive Hydrogel"

_ijms, 2022, doi:10.3390/ijms232213939_

Round 1

Reviewer 1 Report

The study involves preparation of temperature-sensitive hydrogel loaded with IL-1ra to effectively inhibit inflammation associated with periodontitis with diabetes. The study seems well designed and has obtained new findings. However there are few major and minor revisions required and need clarifications from the authors.

1. Aim/objective of the study is to be clearly mentioned in the introduction section.

2. 3rd page, reference no. [8] should appear before full stop.

3. In both FTIR and XRD studies I didn't find FTIR Spectra and XRD patterns of pure IL-1ra. Without IL-1ra spectra, the comparison is not possible. The new peaks seen in XRD spectra may be of IL-1ra. FTIR spectra of individual components and the physical mixture will give information about the incompatibility issues.

4. Figure 2  (d) why authors did not perform XRD studies with beta-GP? 

5. How the authors have determined the cumulative drug release of IL-1ra?

6. What is the composition of simulated body fluid used in drug release studies?

7. No clarity on methodology for degradation and drug release studies.

8. Page 19, Lu's method, RT-qPCR reference to be mentioned

9. Table 3: 'Sequence of probes' to be changed to sequence of primers (5'-3' direction) considering you have not used fluorescent labelled probes but performed syber green assay.

Reviewer 2 Report

This paper entitled “Treatment of periodontal inflammation in diabetic rats with IL-1ra thermosensitive hydrogel” by Liu et al. developed a hydrogel with IL-1ra to treat periodontal inflammation.

The overall quality of this work is very good. The experiment design is good, and proper reference is conduct. There is fair amount of experiment has been performed. The manuscript is well-prepared. Data is convincing and well discussed. Although the experiment design good and data is convincing, there are some of the concerns will be addressed as follows:

Minor Issues

1 It is hard to say it is a thermosensitive hydrogel just because it will form gel at 37oC.

2 Scale bar is missing in figure 3 and figure 6.

3 Limitation of this study can be further discussed.

Round 2

Reviewer 1 Report

The manuscript can now be accepted for publication